# Possible Anti-Aging and Anti-Stress Effects of Long-Term Transcendental Meditation Practice: Differences in Gene Expression, EEG Correlates of Cognitive Function, and Hair Steroids

**DOI:** 10.3390/biom15030317

**Published:** 2025-02-20

**Authors:** Supaya Wenuganen, Kenneth G. Walton, Frederick T. Travis, Tobias Stalder, R. Keith Wallace, Meera Srivastava, John Fagan

**Affiliations:** 1Center for Brain, Cognition and Consciousness, Maharishi International University, Fairfield, IA 52557, USA; ftravis@miu.edu; 2Department of Physiology and Health, Maharishi International University, Fairfield, IA 52557, USA; kwallace@miu.edu (R.K.W.); john.fagan@hrilabs.org (J.F.); 3Institute for Prevention Research, Maharishi International University, Fairfield, IA 52557, USA; 4Department of Psychology, University of Siegen, 57076 Siegen, Germany; tobias.stalder@uni-siegen.de; 5Department of Anatomy, Physiology, and Genetics, Uniformed Services University of the Health Sciences, Bethesda, MD 20814, USA; meera.srivastava@usuhs.edu; 6Health Research Institute, Fairfield, IA 52556, USA

**Keywords:** human gene expression, cortisol, public health, meditation, stress reduction, cognitive function, allostatic load

## Abstract

**Background**: Our previous comparison of peripheral blood mononuclear cells (PBMCs) from long-term Transcendental Meditation^®^ (TM^®^) practitioners and matched non-practitioner controls found 200 differentially expressed (DE) genes. Bioinformatics analyses of these DE genes suggested a reduced risk of diseases associated with stress and aging in the TM group. Here we assessed additional signs of reduced stress and aging. **Methods**: A sample of 15 of the 200 DE genes was studied using qPCR in PBMCs from 40-year TM practitioners (“Old TM”, *n* = 23) compared to a “Young Control” group (*n* = 19) and an “Old Control” group (*n* = 21) of non-meditators. In these three groups, plus a “Young TM”, 12-year practitioner group (*n* = 26), we also studied EEG-based parameters of cognitive function (the Brain Integration Scale (BIS), and latency of three components of the event-related potential (ERP)). Finally, using LC/MS/MS, we compared persistent levels of cortisol (F) and its inactive congener, cortisone (E), in hair. **Results**: qPCR analysis showed that 13 of the 15 genes were more highly expressed in Old Controls than in Young Controls. In the Old TM group, 7 of these 13 were lower than in Old Controls. Both TM groups had higher BIS scores than their age-matched controls. The Old TM group had shorter N2, P3a, and P3b latencies than the Old Control group, and latencies in the Old TM group were not longer than in the Young Control group. The Hair F/Hair E ratio was higher in the control subgroups than in their age-matched TM subgroups, and Hair F was higher in the Young Control and combined control groups than in the Young TM and combined TM groups. **Conclusions**: These results are consistent with reductions in biomarkers of chronic stress and biological age in long-term TM meditators. They are also consistent with results from the previous study suggesting that TM practice lowers energy consumption or leads to more efficient energy metabolism.

## 1. Introduction

Many age-related diseases and disorders involve molecular mechanisms associated with stress [1,2,3,4]. Several recent reviews have attempted to systematize stressor types, different responses to stressors, and the connections of these to diseases and aging [1,5,6]. For the past several decades, the concept of “allostatic load”—i.e., the cumulative total of adaptations to stress that may be damaging to an individual—has proven useful in determining stress-related factors that predispose to disease and aging [7,8,9].

The term “allostasis”, succinctly defined as “stability through change”, is a concept applied to adaptations to potentially stressful internal or external stimuli or environments [7,9,10]. It involves the nervous, endocrine, and immune systems and is more anticipatory than the momentary perturbations characterizing homeostasis [10]. Allostasis requires energy, and allostatic load is associated with an increase in the rate of energy use [11,12]. This may have particular significance for aging, as shown in a recent study of cellular allostatic load [12], and for understanding the wider implications of the current study [13].

Evidence from our recent exploratory study of transcriptional differences in peripheral blood mononuclear cells (PBMCs) in long-term practitioners of Transcendental Meditation (TM) technologies vs. matched non-practitioner controls (i.e., the parent study) suggested that energy use by the TM group was chronically lower or more efficient than energy use by the control group [14]. Because all study participants were selected to be free of known life-threatening diseases, this suggested the demographically matched control group in the genome-wide microarray component of the study was higher in allostatic load. The finding of transcriptional differences in other stress-related pathways and genes in that study supports this conclusion.

Identification of glucocorticoid receptor signaling as one of the canonical pathways arising from pathway analysis of the microarray data in that study is consistent with prior physiologic studies [15,16,17,18]. Together, these studies suggest that reduction of long-lasting effects of stress on the hypothalamic–pituitary–adrenal (HPA) axis, especially reduction of chronically elevated cortisol, is a likely mechanism mediating some of the beneficial effects of TM technologies. Altered function of the HPA axis, particularly alterations leading to chronically elevated cortisol, has long been regarded as a prime biomarker of allostatic load or overload. Recent findings in relation to cardiovascular disease (CVD) [19] and metabolic syndrome (MetS) [20,21] are consistent with this conclusion.

Additionally, based on the genome-wide microarray data in the parent study, it was suggested, and later confirmed [22], that expression of other genes related to the damaging effects of stress, including the conserved transcriptional response to adversity (CTRA) profile, a multi-gene indicator of exposure to chronic threat [23,24], was reduced in this sample of long-term TM practitioners. The CTRA profile, which appears to involve excessive activity of the sympathetic nervous system [25], can be considered another molecular biomarker of allostatic load. It is associated with elevated inflammation and increased susceptibility to infectious diseases [23,24,25]. Increased inflammation may contribute to failure of the immune system to effectively clear pathogens and dysfunctional host cells, causing changes in mitochondrial function and increased abundance of reactive oxygen species, major contributors to aging [26,27].

Cognitive decline is common with advancing age, even in the absence of disease [28,29], and is strongly linked to stress and cortisol levels [3,4,30,31]. In the current study, along with further investigating transcriptional differences that may be due to long-term meditation practice, we evaluated two EEG measures of cognitive function. One of these is the latency of three components of the event-related potential (ERP), i.e., two frontal–central components, N200 (N2, stimulus evaluation) and P300a (P3a, context updating of novel stimuli), and a parietal component, P300b (P3b, context updating of infrequent stimuli—oddballs). The latency of all three components increases with age [32,33] and may increase under conditions of elevated cortisol [34], especially in subjects with poor glycemic control [35,36]. The gradual increase in these 3 latencies after age 20 suggests that this measure is a direct indicator of age-related decline in neural processing speed or efficiency [33,37].

We also examined a comprehensive scale of healthy brain functioning, the Brain Integration Scale (BIS). The BIS includes three EEG measures identified through a stepwise multiple discriminant analysis of EEG measures that distinguished three groups with different levels of expanded experiences: naïve, novice, and expert meditation. These three measures are the brain preparatory response during simple and choice paired reaction-time tasks, broadband frontal EEG coherence, and alpha/gamma power ratios during a vigilance task [38,39]. The BIS has been reported to relate to higher performance in non-meditators. Examples are: world-class athletes [40] and business leaders [41], as well as correlations with faster habituation to stressful stimuli in college students [39]. BIS levels also correlated positively with creativity in a study of product development engineers [42].

As a long-term indicator of endocrine consequences of chronic stress that may reflect allostatic load [19,43,44], the concentrations of the glucocorticoids cortisol (F) and cortisone (E) in hair were also measured in all subjects. F, the active glucocorticoid, and E, the inactive form, are enzymatically interconverted both in specific tissues and systemically [45,46]. The concentrations in hair of both F (Hair F) and E (Hair E) correlate with the risk of MetS [20], a condition previously reported to be reduced by TM practice [47].

The microarray analysis in the parent study focused on a comparison of a small group of long-term (mean, 38 years) practitioners of TM technologies with demographically well-matched controls [14]. The present study focuses on two larger, less well-matched groups of long-term (means of 12 and 40 years) TM practitioners and age-matched controls. We hypothesized that these comparisons also would show signs consistent with anti-stress and anti-aging effects of meditation, namely, 1) altered expression of specific genes related to aging or stress, 2) beneficial effects on cognitive function, as indicated by EEG tests, and 3) signs of lower systemic cortisol or better adaptation of HPA axis function.

## 2. Materials and Methods

### 2.1. Research Design and Participants

This cross-sectional, observational study compared the following 4 groups of approximately 25 subjects each: 1. Young Control, 2. Young TM, 3. Old Control, and 4. Old TM. An overview of subject characteristics and N for each of the three approaches is presented in Table 1. Additional demographics for EEG subjects are available in Table 2. The research design and methods were approved by the Institutional Review Board of Maharishi International University. Following both written and oral explanations of the study, participants gave signed written consent prior to participation.

Participants from the two age ranges (“Young”, aged 20–30; “Old”, aged 55–72) were recruited through distribution of advertising brochures on the campus of Maharishi International University and in other public places within a 40-mile radius of Fairfield, Iowa. Interested individuals were prescreened based on a questionnaire regarding their health status and other demographic and lifestyle variables. Note that all qPCR runs were performed at one time, i.e., as part of the original study, and that the EEG studies were performed in the same timeframe and the same subjects as the blood sampling for transcriptomic comparisons. qPCR data for the Old TM group and the Old Control group were presented in the original study [14]. For comparison with the previously unpublished results in the Young Control group, those data are presented in a different format in the second figure of this study. Note also that individuals in the Old TM group averaged having practiced the TM technique for 39.5 ± 2.8 years whereas those in the Young TM group averaged 12.2 ± 5.4 years. In most contexts, both groups would be considered long-term practitioners. Due to restricted resources, qPCR analyses were not performed in the Young TM group.

Both genders were accepted into the study. To minimize genetic variation, only one ethnic group (Caucasian) was admitted. Prospective participants were excluded if they reported having had a doctor-identified history of diabetes, nerve damage, heart attack, coronary heart disease, stroke, kidney failure, cancer, any other life-threatening illness, major psychiatric disorder, or substance abuse. Candidates for control participation were excluded if they had ever been instructed in the TM program. Practitioners of the TM technique, with or without the advancement known as the “TM-Sidhi program”, were excluded if they did not report being “regular” in their practice (i.e., usually twice a day).

### 2.2. Procedures

#### 2.2.1. PBMC Preparation, RNA Extraction, Measurement, and Integrity Check

Blood samples were collected from the Young Control group, the Old Control group, and the Old TM group as described in our previous publication [14]. Likewise, the methods of RNA extraction, measurement, and integrity determination were the same as before, as was the qPCR methodology. In brief, the buffy coat was isolated using BD Vacutainer^®^ CPT^TM^ Mononuclear Cell Preparation Tubes (Franklin Lakes, NJ, USA). The supernatant and the cell pellet were then separated and stored at −80 °C. RNA was extracted from the buffy coat using the RNA-Bee^TM^ Kit (Tel-Test, INC, Friendswood, TX, USA), with concentration estimated using a UV absorption ratio method [48]. Ratios above 1.7 were considered sufficiently pure. RNA integrity was analyzed by automated electrophoresis on microfluidic labchips using the Experion^TM^ System (Bio-Rad, Hercules, CA, USA), with acceptable cutoff values being in the range 7 < RQI ≤ 10. Samples meeting the criterion for integrity were then ready for use as input for cDNA labeling employing the MessageAmp^TM^ II Kit (Ambion® by Life Technologies, Carlsbad, CA, USA).

### 2.2.2. qPCR Methodology

Target genes for comparison among the different groups were chosen more or less randomly from the DE genes identified in our previous study of global gene expression [14]. The list and design of primers, construction of cDNAs, and qPCR procedures are all detailed in the previous paper [14].

#### 2.2.3. EEG Recording

##### Event-Related Potentials

EEG was recorded from 32 active sensors in the 10-10 system using the BIOSEMI Two amplifier and acquisition software system (ActiView version 9.02, Amsterdam, The Netherlands) [49,50,51]. Two additional sensors were applied to the left and right earlobes for re-referencing offline [39]. One sensor was attached on the non-dominant wrist for measuring heart rate [52], and a respiration belt was applied around the waist to measure breathing rate. All signals were digitized at 256 samples/s with no high- or low-frequency cutoff, and were stored for later analysis offline. Table 2 contains demographic information most relevant to EEG recording participants.

An oddball auditory event-related potential task with rare unexpected and rare expected targets was used to elicit ERPs [53]. Subjects received 240 stimuli with 1 sec inter-stimuli intervals. Twenty-four of the stimuli (10%) were rare, unexpected, 85 dB white noise bursts (*novel* stimuli). Twenty-four of the stimuli (10%) were rare, expected, 85 dB 1000 Hz tones (*oddball* stimuli). The other 80% of the stimuli (192) were standard, 85 dB 500 Hz tones, which subjects were told to ignore.

##### Recording Continuous EEG to Calculate the Brain Integration Scale

EEG was also recorded while subjects performed two paired reaction-time tasks. The first was a 2 min simple reaction-time task when participants were presented an asterisk on a computer screen (S1) that was followed 1.5 s later by a continuous computer-generated tone (S2, 1200 Hz, 85 dB). They were asked to press the space bar as soon as they heard the tone. The second was a 2 min choice reaction-time task when participants were presented a whole number on a computer screen (S1) that was followed 1.5 s later by another whole number. They were asked to press a right- or left-hand button to indicate which of the two numbers was larger in value.

#### 2.2.4. Hair Steroid Concentrations

Following the protocol described in Gao et al. [54], 4 steroids (F, E, testosterone, and progesterone) were present in sufficient quantities for analysis in most samples. In brief, bundles of hair strands (~3 mm diameter) were cut as close as possible to the scalp from a posterior vertex position using fine scissors. The proximal 3 cm hair segment was used for analyses. Based on an average hair growth rate of 1 cm/month [55], this segment is assumed to reflect a period of about 3 months prior to sampling. In brief, samples were washed in 2.5 mL isopropanol for 3 min and cortisol was extracted from 7.5 mg of whole, non-pulverized hair using 1.6 mL methanol in the presence of 20 μL cortisol-d_4_ as an internal standard for 18 h at room temperature. Samples were centrifuged at 15,200× *g* relative centrifugal force for 2 min, and 1 mL of the clear supernatant was transferred into a new 2 mL tube. The alcohol was evaporated at 50 °C under a constant stream of nitrogen and reconstituted with 175 μL double-distilled water. Samples were analyzed using a Shimadzu HPLC–tandem mass spectrometry system (Shimadzu, Canby, OR, USA) coupled to an AB Sciex API 5000 Turbo-ion-spray triple quadrupole tandem mass spectrometer (AB Sciex, Foster City, CA, USA) with purification by online solid-phase extraction. This method has been shown to achieve excellent sensitivity, specificity, and reliability (intra- and inter-assay CVs between 4.7 and 8.8%) [54] and is considered the current “gold standard” for hair steroid analysis [56].

### 2.3. Data and Statistical Analysis

#### 2.3.1. qPCR Ct (Threshold Cycle) Values

*qPCR* threshold cycle (Ct) values were normalized to generate ∆Ct by subtracting the Ct value of the target genes from the Ct value of the reference gene, using the BioRad CFX Maestro™ software 2.1 intended for this purpose. Then, the average of the ∆Ct of each group was calculated to obtain the ∆∆Ct for the expression-rate comparison of the Old Control group vs. the Old TM group and the Young Control group vs. the Old Control group. The expression rate of each gene was calculated using a formula (2^(∆∆Ct)^). Then *p* values were determined using SPSS 13.0 (Chicago, IL, USA). Outliers were adjusted using the Winsorization procedure, which replaces the outlier with the next closest value, and then kurtosis and skewness < 1 for normal distribution were observed.

Next, MANOVA tests were conducted for the Old Control group vs. the Old TM group to find the meditation effect, the Young Control group vs. the Old Control group to find the aging effect, and the Young Control group vs. the Old TM group to find the anti-aging effect. An alpha level of *p* ≤ 0.05 was adopted for statistical significance and *p* < 0.1 for statistical trend.

#### 2.3.2. Event-Related Potentials (ERPs)

The raw data records were manually scanned for artifacts, and artifacts were removed from the analyses. The remaining data were analyzed with the BrainVision Analyzer 1.3 (Munich, Germany) (http://www.brainproducts.com/downloads.php?kid=9, accessed on 29 March 2013). Data were binned by onset of the novel and oddball stimuli in 900 ms windows—100 ms before stimulus onset and 800 ms after the stimulus [53]. The amplitude and latency were recorded for the N2 (stimulus evaluation in a 200–350 ms window in frontal–central sensors), P3a (context updating to novel stimuli in a 300–500 ms window in frontal–central sensors), and P3b (context updating to rare, expected stimuli in a 300–500 ms window in the parietal sensors).

#### 2.3.3. Brain Integration Scale (BIS)

To calculate the brain preparatory response, 2 s epochs were extracted from the data stream beginning 100 ms pre-S1 and ending 400 ms post-S2 during both the simple and choice reaction-time tasks. For both the simple and the choice trials, the late CNV was measured in microvolts as the average amplitude in the 200 ms window before the second stimulus, relative to the 100 ms baseline. Simple-choice difference scores were calculated (CNVsimple minus CNVchoice) to assess the impact of the additional cognitive load of the choice trials independent of possible group differences in the simple trials.

Power and coherence estimates were calculated while the subjects performed Connor’s Performance Task—Identical Pairs (CPT-IP), a measure of frontal executive functioning. Participants were presented a capital letter every 0.9 s for 2 min and were instructed to press the left button every time a letter appeared that was different from the preceding one. They were instructed to press the right-hand button every time the current letter was the same as the preceding letter. This task is challenging because the letters come quickly, and 80% of the letters require a left-hand button press. Participants develop a response bias for left-hand responses. When the rare right-hand response is required, the frontal executive system has to inhibit the left-hand response and execute the correct response.

Data from the CPT-IP were visually scanned for epochs with physical movement, including electrode and eye-movement artifacts. These were manually marked and were not included in the spectral analysis. The artifact-free data were then digitally filtered with a 2–50 Hz band pass filter with 48 dB roll off and fast-Fourier-transformed in 2 s epochs. Absolute power (µV^2^/Hz) was calculated from 2–50 Hz in alpha, beta, and gamma bands at the 32 recording sites. Coherence was calculated for the 496 possible combination pairs of 32 recording sites in the same 3 bands. EEG coherence was taken as the absolute value of the cross-correlation function in the frequency domain, reflecting the number and strength of connections between spatially distant brain areas.

The values of broadband frontal coherence, the alpha/beta absolute power ratios from the CPT-IP, and the CNV difference scores were converted to z-scores and compared to the values in the normative database from our earlier studies [38,39]. This database included non-TM, short-term TM, and long-term TM participants. To prevent any subgroup from having a mean BIS value below zero, a value of 2 was added to the score of each subject.

Final analyses of the three ERP latencies and the BIS values were conducted with an omnibus MANOVA with age and TM status as between factors. Individual ANOVAs were used to test individual contrasts. An alpha level of *p* ≤ 0.05 was adopted for statistical significance.

#### 2.3.4. Hair Steroids

In the statistical analysis of hair steroids, the overall pattern of glucocorticoid data was similar to that observed in a previous study of 1258 employees of a large German aerospace company, and a similar statistical approach was used [20]. Both Hair F and Hair E data were skewed to the positive side. Therefore, data were tested in raw form as well as log_10_ transformed in parametric analyses. Two subjects with extremely high Hair F values (>3 SD from the mean) were removed unless otherwise stated. In paired comparisons of TM and Control groups when the homogeneity of variances assumption was not met, either the unpaired *t*-test with separate variance estimates or the non-parametric Mann–Whitney U test was used. An alpha level of *p* ≤ 0.05 was adopted for statistical significance.

## 3. Results

### 3.1. Comparisons of Gene Expression Using qPCR

Relative expression of 15 genes randomly selected from the 200 DE genes found in the genome-wide parent study was compared in three groups, namely, the Young Control, Old Control, and Old TM groups. Most of the genes (13 of the 15) were up-regulated in the Old Controls compared to the Young Controls (Figure 1; a positive gene expression ratio indicates up-regulation relative to the comparison group). 

In contrast, expression of these genes in the Old TM group compared to the Old Control group (Figure 2) showed that only 1 of the genes (*CXCL10*) was significantly up-regulated in the TM group while 7 of 15 were down-regulated (3 significantly, 4 trending). Six were highly expressed in the Old Control group compared to the Young Control group and were considered likely age-associated genes. Ct data for subjects in each group are available in Appendix A.

Figure 3 shows the relative expression patterns among the three groups, as indicated by threshold cycle values (Ct), for the six DE genes that appeared to be age-associated. Except for *SOCS3*, expression in the Old TM group was intermediate between that of the Young Control group and the Old Control group. Expression of *SOCS3*, *ITGB5*, and *TAL1* in the Old TM group was statistically the same as in the Young Control group (see Table 3). Expression of two other genes, *ITGB3* and *LMNA*, in the Old TM group was significantly higher than in the Young Control group, and expression of *ALOX12* trended higher (Figure 3).

### 3.2. Comparison of N2, P3a, and P3b Latencies

An omnibus MANOVA conducted with two between factors (Age Group and TM Status) and N2, P3a, and P3b latencies as variates resulted in highly significant main effects for both factors, with latencies for the Young Group (collapsing across TM Status) and the TM Group (collapsing across Age Group) being shorter. The interaction was not significant (*F*(3, 61) = 1.151, *p* = 0.287). Table 2 (in Section 2) lists demographic data of special importance for EEG data collection. Table 4 presents the N in each group, mean (SEM) of latencies of the three components, *F*-statistics, and effect sizes. ERP and BIS data for each subject are available in Appendix A: Brain Integration Scale, Grouping Factors, and N2, P3a, and P3b Latencies.

Figure 4 and Figure 5 show the grand averages ± SD of the ERP recordings for the novel and oddball stimuli at the appropriate recording sites. Figure 4 shows the tracings for N2 and P3a from novel stimuli. Figure 5 shows the tracings with P3b from oddball stimuli.

One-way ANOVAs were used to compare the latencies of the three components in two subgroup comparisons: Old TM vs. Old Control and Old TM vs. Young Control. Table 5 presents the results of this analysis. The Old TM group had significantly shorter latencies than those of the Old Control group. The N2 and P3b latencies were not significantly different from those of the Young Control group. P3a latencies were shorter in the Young Control group.

### 3.3. Comparison of BIS Scores

BIS scores were determined for the three groups. Table 6 shows results of the initial two-factor omnibus ANOVA comparison of scores with the two categorical variables being TM Status (TM, Control) and Age Group (Young, Old). There were significant group effects for higher BIS scores for TM Status, collapsing across Age Group, and the younger group, collapsing across TM Status. There were no significant interactions. Individual BIS scores are available in Appendix A: Brain Integration Scale, Grouping Factors, and N2, P3a, and P3b Latencies.

Table 7 shows the subgroup ANOVA comparisons of BIS scores for the Old TM–Old Control and the Old TM–Young Control comparisons. As seen in the table, the mean BIS score for the Old TM group was significantly higher than for the Old Control group and not significantly different from that of the Young Control group.

Within the separate young and old groups, correlations between BIS and chronological age were not significant. This can be seen in the long-term Old TM group, for example, where the small negative correlation between BIS and chronological age was far from significance: *r*(26) = −0.065, *p* = 0.751. In this same subgroup, the correlation of BIS with months of TM was positive but fell short of significance: *r*(26) = 0.256, *p* = 0.207.

Exploratory comparison of ERP latencies and BIS scores for the Young TM versus the Young Control showed no significant differences between the two younger groups in ERP latencies (*F*(3,31) = 1.9, *p* = 0.15, but the mean BIS score for the Young TM group (5.3 ± 0.3) was significantly higher than for the Young Control group (4.3 ± 0.4) (*F*(1,49) = 4.5, *p* = 0.038).

### 3.4. Comparison of Hair Glucocorticoid Concentrations

Hair F values were significantly lower for the combined long-term TM practitioner subgroups (Total TM group) than for the combined Control subgroups (Total Control group) and for the Young TM group than for the Young Control group but were not significantly lower for the Old TM vs. Old Control comparison. The Total TM value was (M ± SE) 3.22 ± 0.466 pg/mg (min = 0.46, max = 22.3) while that of Total Control was 8.61 ± 2.50 pg/mg (min = 0.60, max = 114.6), *t*_separate variances_ (53.5) = 2.132, *p* < 0.04, Glass’s *delta* = 0.30. In the Young TM group, Hair F was 2.88 ± 0.457 pg/mg (min = 0.46, max = 11.16) while in the Young Control group it was 7.64 ± 2.11 pg/mg (min = 0.60, max = 42.0), *t*_separate variances_ (30.6) = 2.19, *p* < 0.04, Glass’s *delta* = 0.42. Hair F in the Old TM group was 3.59 ± 0.783 pg/mg, (min = 0.67, max = 22.3), while for the Old Control group it was 9.88 ± 5.09 pg/mg (min = 0.72, max = 114.6), *t*_separate variances_ (22.0) = 1.23, *p* = 0.23, Glass’s *delta* = 0.27. Hair F values (see above) for the Old TM group trended lower than for the Young Control group, *t*_separate variances_ (35.5) = 1.82, *p* = 0.077, Glass’s *delta* = 0.36. Data for individual subjects are shown in Appendix A: Hair Steroids, Grouping Factors, and Gene Expression Ct Values.

Raw Hair E values did not significantly differ between long-term TM practitioner groups and the age-matched Control groups. Hair E in the Old TM group was (M ± SE) 8.72 ± 1.33 pg/mg, (min = 1.05, max = 37.08), while for the Old Control group it was 10.11 ± 1.97 pg/mg (min = 0.45, max = 30.06). In the Young TM group, Hair E was 8.36 ± 0.98 pg/mg (min = 1.94, max = 23.2) while in the Young Control group it was 7.71 ± 1.26 pg/mg (min = 1.95, max = 38.64). Hair E values for the young and old groups combined were: Total TM = 8.56 ± 0.839 pg/mg (min = 1.05, max = 37.1), Total Control = 8.74 ± 1.12 pg/mg (min = 0.45, max = 38.6).

Because of the enzymatic interconversion (also known as “recycling”) of F and E, we reasoned that the Hair F/Hair E ratio might more accurately reflect glucocorticoid status than either steroid alone. Figure 6 shows comparisons of the mean Hair F/Hair E ratios.

These results for ratio differences were similar to those for Hair F alone, but they appeared to show more robust differences and gave greater effect sizes. In an attempt to better understand the quantitative relationship between Hair F and Hair E, we conducted an analysis of regression of the log_10_ values of dependent variable Hair F on the explanatory variable log_10_ Hair E, examining all cases with no missing data (N = 108, after exclusion of two subjects with missing values). Figure 7 shows the scatter plot of the data with the ordinary least squares (OLS) regression line. The estimate of the slope regression coefficient for the explanatory variable log_10_ E is 1.2389 (*t*(106) = 9.56, *p* < 0.001), with beta coefficient 0.6959. R-square is 0.4631, with adjusted R-square 0.4581, indicating that log_10_ E explains approximately 46% of the variation in log_10_ F. The analysis was performed with Stata 18 software (StataCorp LLC, College Station, TX, USA, 2023).

However, a formal analysis (Doornik–Hansen test) rejected the null hypothesis that the regression errors are drawn from a normal (Gaussian) distribution (*p* < 0.001). This finding violates a key assumption of OLS regression. Formal testing [57] also indicated violation of the OLS assumption of no heteroskedasticity of the regression errors (*p* < 0.001). Heteroskedasticity is defined as non-constant conditional variance of the regression errors given the predictor variables.

The rejection of normal regression errors and visual inspection of Figure 7 suggest the possible presence of a few extreme or outlying positive observations. These potential outliers may simply reflect inherent variability of log_10_ F or may indicate that at least some of the outliers are observations drawn from a different data distribution.

Re-analysis after sequential exclusion of 12 positive outliers and 1 negative outlier by removing observations that were more than 3.5 standard errors from the regression line yielded a data set that meets all assumptions for OLS regression. After deleting these postulated outliers, the estimated slope coefficient decreased to 1.0836 (*t*(94) = 19.750, *p* < 0.001). The R-square was 0.9132, with adjusted R-square 0.9012, indicating that log_10_ E explained approximately 90% of the variation in log_10_ F.

To further characterize this high-Hair-F “Positive Outlier” subgroup, comparisons were made with the remaining “Ordinary” subgroup. Inspection of the raw data confirmed that all members of the Positive Outlier subgroup showed Hair F/Hair E ratios greater than 1.0. All members of the Ordinary group showed Hair F/Hair E ratios less than 1.0.

Because this ratio is a measure of the balance between the active glucocorticoid F and its inactive congener E, we further explored these two groups by comparing their differences in the main variables using the non-parametric Mann–Whitney U test (see Table 8). Mean Hair F in the Positive Outlier group was 8.9 times higher than in the Ordinary group. Mean Hair E in the Positive Outlier group, while numerically 2.3 times higher than that of the mean for the Ordinary group, was not statistically different. The Hair F/Hair E ratio showed the greatest difference between groups (12-fold).

The number of cases falling in the Positive Outlier group differed between TM and Control subjects. Two of the Old TM cases and none of the Young TM cases were in the Positive Outlier group. The 10 Control cases in the Positive Outlier group fell equally into Young Control and Old Control subgroups. Overall, five male and seven female cases fell into the Positive Outlier group.

In the Ordinary group, Hair F, Hair E, and the Hair F/Hair E ratio were compared between TM and Control groups. No significant differences in Hair F or Hair E were found between Total TM and Total Control in the Ordinary group: Hair F: *t*(94) = 0.631, *p* = 0.53; Hair E: *t*(94) = 0.022, *p* = 0.98. For the Hair F/Hair E ratio in the Ordinary group, there was a numerical tendency for lower mean (± SE) in the Total TM subgroup (0.3399 ± 0.020 pg/mg) compared to the Total Control subgroup (0.3873 ± 0.024 pg/mg), *t*(94) = 1.54, *p* = 0.13.

## 4. Discussion

The main findings of the study appear to support the stated hypotheses. These findings are: (relating to hypothesis 1) lower expression of 13 of the 15 selected genes in the Young Control group than in the Old Control group; lower expression of 7 of these 15 in the Old TM group than in the Old Control group; identification of specific DE genes that appear to reflect an anti-aging, anti-stress state in the Old TM group, (relating to hypothesis 2) shorter latencies of N2, P3a, and P3b (indicating higher cognitive processing speed) and higher BIS score (indicating higher cognitive performance) in the TM groups, and (relating to hypothesis 3) lower Hair F/Hair E ratio in TM groups in all comparisons of TM and control, along with lower mean Hair F in three of the four group comparisons. These outcomes are discussed in detail in the following subsections because of their relevance not only to effects of this meditation practice but also because they may help lay groundwork for new biomarkers of stress and aging.

### 4.1. Aging-Related Differences in Gene Expression

Participants in the present study were more numerous (*n* = 23 and 21 for the Old TM and Old Control groups, respectively) and were not as well matched as in the microarray comparison [14]. Nevertheless, qPCR analysis in this more diverse group generally validated the microarray findings. Furthermore, in the present analysis, 13 of the 15 genes that were randomly selected from the DE genes in that microarray also showed an apparent age-related difference, i.e., all 13 of these genes were more expressed in the Old Control group vs. the Young Control group. Six of these showed lower expression in the Old TM group than in the Old Control group, suggesting possible anti-aging and/or anti-stress effects of long-term meditation practice. Other researchers report that, among the small percentage of genes that differ with chronologic age, most are overexpressed in older age groups [58,59,60]. In one large-scale transcriptomic study, for example, 2% of the probes were associated with age; 82% of these were overexpressed in the older group [59].

### 4.2. Differential Expression of Specific Genes Indicating Anti-Aging and Anti-Stress Effects

The first four of the six genes identified here as reflecting possible anti-aging, anti-stress effects of TM (*SOCS3*, *ITGB3*, *ITGB5*, *LMNA*, *TAL1*, and *ALOX12*) were also reported by others to show age-related increases in expression in PBMCs [60]. The following subsections describe their known functions and other evidence associating them with aging and stress.

#### 4.2.1. *SOCS3* Expression—Chronic Stress, Energy Metabolism, and CVD

Among the 15 genes studied, *SOCS3* expression provides the clearest links to possible anti-stress and anti-aging effects. Expression of *SOCS3* in the Old TM group was significantly down-regulated compared to the Old Control group and somewhat lower compared to the Young Control group.

Suppressor of cytokine signaling 3 (SOCS3) is a signaling protein induced by pro-inflammatory cytokines, including IL-6 and IL-10, and is increased in the stress response [61]. SOCS3 plays a major role in regulating inflammation and infections [62,63]. SOCS3 also plays a vital role in a complex of energy regulatory mechanisms involving interactions with AMPK, JAK-STAT, and leptin [64]. In both these roles, *SOCS3* down-regulation is consistent with a healthier state.

The SOCS3 effect on AMPK is particularly important. AMPK is an energy-status sensor for maintaining energy homeostasis [65,66]. Its full activation is prevented by SOCS3 [64]. Not only does AMPK directly affect energy metabolism through its influence on oxidative phosphorylation, but it also regulates mitochondrial generation and disposal, being responsible for ensuring there are enough functional mitochondria for normal energy metabolism [65]. Elevated level of SOCS3, by inhibiting full activation of AMPK, interferes with mitochondrial energy metabolism [64]. The parent study [14] provided evidence for reduced erythropoiesis in a long-term TM group compared with the matched control group, consistent with reduced or more efficient energy metabolism in the TM group, that could derive from reduced SOCS3.

Increased level of SOCS3, then, appears to provide a key molecular mechanism linking chronic stress to reduced mitochondrial function and aging. This is consistent with the recent recognition of mitochondrial energy metabolism as a key aspect of stress responses and adaptation [11] and may explain mitochondrial sensitivity to chronic stress. Decreased mitochondrial number and function due to chronic stress are known to contribute to aging [67]. In addition, mitochondrial health is sensitive to mood and caregiver stress [68], an effect modifiable by effective stress-reduction approaches.

A recent study of cellular allostatic load and energy metabolism in fibroblast cell lines may reinforce this interpretation of the *SOCS3* results and also connects importantly with our results on glucocorticoid activity, i.e., the Hair F/Hair E ratio, in this paper. Bobba-Alves et al. [12] studied effects of growing fibroblasts in a medium with elevated glucocorticoid concentrations comparable to those found in an organism with high allostatic load. This chronic glucocorticoid exposure caused a 60% increase in energy use in the fibroblasts along with a shift from glycolysis to oxidative phosphorylation. These signs of increased energy metabolism were accompanied by multiple signs of increased cellular aging [12]. In the present study, increased systemic glucocorticoid, as reflected by increased Hair F and increased Hair F/Hair E ratio, provides a parallel situation and probable explanation for the elevated oxygen demand evidenced in the control group in the parent study [14].

Clinical trials have found TM practice to prevent or reverse insulin resistance [47], a precursor to diabetes and obesity. Increased SOCS3 level mediates the inhibitory effects of IL-6 on insulin signaling and glucose metabolism and is documented to lead to insulin resistance in skeletal muscle, liver, and adipose tissue [64]. Other researchers have verified that chronic JAK-STAT3-SOCS3 signaling induced by leptin and IL-6 is implicated in obesity [69,70], cancer [71], and aging [60,72].

This reduced expression of *SOCS3* aligns with other research on effects of the TM program. Specifically, randomized controlled trials (RCTs) indicate this program reduces insulin resistance [47], atherosclerosis [73], other risk factors for CVD (see, for reviews, [74,75]), and cardiac events [76]. Studies also suggest practice of the technique has beneficial effects in diabetes [47,77]. Evidence for novel effects of the TM technique on energy metabolism was among the earliest findings of research on this technique (see, for review, [78]), and *SOCS3* expression provides a possible mechanistic link.

#### 4.2.2. Expression of *ITGB5* and *ITGB3*—Inflammaging, Platelet Aggregation

Two members of the integrin family of genes that were studied in this qPCR analysis (*ITGB5* and *ITGB3*), and another member (*ITGA2B*) observed in our microarray comparison [14], were down-regulated in the Old TM group compared with the Old Control group. The integrin beta 5 (ITGB5) gene product, integrin, is important in platelet aggregation. Its related super-pathways are integrin-mediated cell adhesion and extracellular matrix–receptor interaction [79,80]. ITGB5 also is associated with extracellular matrix stiffness, one of the hallmarks of aging [81].

In addition to their adhesive functions, integrins can activate intracellular signaling pathways that control growth, differentiation, apoptosis, cell motility, cell migration, and cell survival [79,80,82]. Up-regulation of these genes is associated with increased production of inflammatory mediators through their roles as signaling molecules (e.g., IL-8 signaling, tight-junction signaling, and ILK signaling) [60]. Other researchers, studying the up-regulation of *ITGB3* with age, have suggested that the integrin β3 subunit is a marker for and regulator of cell senescence [83].

Besides inflammation, elevated expression of *ITGB* genes increases cell-to-cell aggregation and cell-to-extracellular matrix adhesion. This characteristic can lead to low mobility of cells in the circulatory system. Increased cell aggregation and adhesion in the blood contributes to disorders such as atherosclerosis, heart attack, and stroke, which are more common in old age [84]. Recently, integrins have been mentioned as novel therapeutic targets for treating cardiovascular disease [85].

#### 4.2.3. *TAL1* and *ALOX12* Expression—Links to Leukemia, Telomerase Activity, and Oxidative Stress

Expression levels of *TAL1* and *ALOX12* were low in the Young Control group and in the Old TM group but higher in the Old Control group than in either of these. T-cell acute lymphoblastic leukemia 1 (*TAL1*, also known as “stem cell leukemia” or *SCL*) is a protein-coding gene that encodes a transcription factor regulating both embryonic and adult hematopoiesis [86]. In human adults, *TAL1* is up-regulated in erythropoiesis and down-modulated in granulopoiesis [87]. Increased level of TAL1 leads to hypersensitivity to erythropoietin, resulting in excessive erythrocytosis [88], and is the most common molecular abnormality found in human T-cell leukemia [89,90]. The elevated *TAL1* expression found here in the Old Control group is consistent with the mechanism described previously (i.e., increased erythropoiesis, see [14]) that may help compensate for the loss of energy efficiency due to elevated SOCS3. TAL1/SCL is also known to be important in vascular development and inflammation [91].

The possible anti-aging effect of reduced expression of *TAL1* suggested by our data may also relate in part to the role of TAL1 as a negative regulator of the promoter of hTERT, the protein subunit of the human telomerase gene. Because increased TAL1 leads to a decrease in hTERT mRNA abundance and hence to reduced telomerase activity [92], which is expected to accelerate aging [93], the relative down-regulation of *TAL1* in the TM group is consistent with an anti-aging effect.

*ALOX12* codes for the enzyme arachidonate 12-lipoxygenase, which metabolizes arachidonic acid to generate potent inflammatory mediators and plays an important role in inflammation-associated diseases. ALOX12 and inducible nitric oxide synthase (iNOS) are well-known mediators of inflammation [94] and oxidative stress [95] in advancing age. Furthermore, ALOX12 has an important function in obesity-related complex phenotypes, including hypertension [96], atherosclerosis [97], diabetes and insulin secretion [98], and obesity itself [99].

#### 4.2.4. *LMNA* Expression—Chromatin Structure and Nuclear Stability

*LMNA*, coding for the lamin proteins, is involved in nuclear stability and chromatin structure. Inflammation appears to be a part of the accelerated aging phenotype caused by *LMNA* mutations [100]. Additional reports have suggested that reduced *LMNA* expression is involved in healthy human aging [101,102]. Others have reported up-regulated expression of this gene in old age in a large meta-analysis [58] and in a study of PBMCs in nonagenarians [60].

To summarize the data on reduced gene expression in the TM group, the association of these six genes with healthy aging through their roles in controlling inflammation, energy metabolism and mitochondrial function, stability of nuclear DNA, and other key cell functions is clear. Increased expression of these genes is connected with a number of age-related diseases. Other transcriptomic studies of PBMCs and muscle are consistent with our findings concerning the key pathways that increase with age and may reflect a common aging signature [103]. The age-related down-regulation of adult cell proliferation pathways, e.g., through TAL1, is a natural strategy to prevent unregulated hematopoietic cell growth and for maintenance of telomerase activity. A separate study of hypertensive patients reported that the TM program increased telomerase gene expression [104]. Other techniques of meditation also may increase telomerase activity [15,105].

It should be noted that in certain tissues other than blood, up-regulation (instead of the down-regulation found in PBMCs) of one or more of these six genes with age may represent a healthier state. For example, in brain tissue, integrin signaling (controlled by *ITGB* genes) is needed because of its important roles in modulating neuronal survival by protecting against oxidative stress and apoptosis [106] as well as in neurogenesis, axonal outgrowth, and axonal pathfinding [107]. Also, continual *LMNA* expression is needed in many tissues for the integrity of nuclear DNA [102].

#### 4.2.5. Up-Regulation of *CXCL10*

The *CXCL10* gene codes for C-X-C motif chemokine 10, also known as interferon gamma-induced protein 10 (IP-10) or small-inducible cytokine B10, a small cytokine belonging to the CXC chemokine family. In the current study, it is the one gene up-regulated in the Old TM group relative to the Old Control group. The significance of this difference is unclear without more information. Up-regulation of *CXCL10,* in conjunction with a unique structural domain, Glu-Leu-Arg, within the protein, can have tumor suppressor and tissue repair roles rather than pro-inflammatory and tumorigenic roles. This unique structural domain allows it to attenuate angiogenesis and have anti-tumor action [108]. An in vitro study found that a considerably higher concentration of the specific variant of CXCL10 with this structural domain is required to exhibit anti-inflammatory and anti-tumor actions [109]. If this variant is present in the older TM group, then the up-regulation of CXCL10 would have tumor suppressor actions. Further studies are required.

### 4.3. Latency of N2, P3a, and P3b

Our data indicate that the N2, P3a, and P3b latencies for the Young TM and Young Control collapsed together were shorter than for the Old TM and Old Control subjects collapsed together. This is consistent with a major association of latency with age. Further, the N2 and the P3b latencies of the Old TM group were not significantly different than those in the Young Control and significantly shorter than those in the Old Control subjects, consistent with possible anti-aging effect of this meditation technique.

N2, P3a, and P3b latencies index different steps of cognitive processing. The N2 component indexes response inhibition during stimulus evaluation—irrelevant stimuli are dampened so that the object in attention can be processed [110]. The P3a originates from stimulus-driven disruption of frontal attention engagement during task processing [111]; this occurred in this study when the novel white noise stimulus interrupted the counting task. The P3b originates when parietal mechanisms take the result from the stimulus evaluation processes and update the context [112]; this occurred in this study when subjects updated the mental sum of the number of oddball stimuli they saw. These three components contribute to the relative timing of cognitive processing [113].

These findings of the effect of age on N2, P3a, and P3b latencies are consistent with results of other studies and with our gene expression data. For example, the oxidative stress-related gene *ALOX12* is down-regulated in the Old TM group; oxidative stress as well as ALOX12 are associated with reduced cerebral cortical thickness [114]. These findings are also consistent with shorter P3 latencies in both auditory and visual oddball tests for long-term meditation practitioners compared to non-practitioner controls [115,116].

The latency of these components gives a cortical index of the maintenance of a more youthful cognitive processing speed with aging. Further supporting this evidence of improved cognitive function in the TM group, a random-assignment, prospective study on aging in individuals in their 80s under residential care found marked improvements in several measures of cognitive function as well as in longevity after beginning TM practice [117].

### 4.4. BIS Score

The BIS scores in the Old TM group were significantly higher than those in the Old Control subjects and not significantly different than those in the Young Control subjects. This suggests that Transcendental Meditation practice may help to maintain higher levels of brain integration as we age.

Previous studies of BIS support the significance of BIS scores for cognitive function. Groups of non-meditators at different levels of achievement exhibited higher BIS scores in mid-level managers [41] and in winning athletes [40] relative to those who seldom placed at the top. In other studies, the BIS was found to correlate positively with moral reasoning, inner directedness, and emotional stability and to correlate negatively with anxiety [118]. Prospective, randomized, controlled trials provide direct evidence that introducing TM practice into one’s daily routine can increase BIS scores after as few as 10 weeks of 20 min twice-daily practice [39,119]. Higher BIS scores, along with shorter N2, P3a, and P3b latencies, support the possibility that TM practice leads to a more youthful state of brain function into older chronological age.

### 4.5. Hair Glucocorticoid Concentrations

Hair F/Hair E ratios were significantly higher in controls in all comparisons of control subgroups with TM subgroups. Hair F alone also was higher in all control subgroup comparisons except for the Old Control vs. Old TM subgroup, which fell short of significance. Close examination of the hair steroid results revealed that most of the difference between TM and Control groups was due to a small subgroup with extremely high Hair F and Hair F/Hair E ratio. This “Positive Outlier” subgroup represented 19% of all control subjects but only 3.6% of TM subjects. It was characterized by a 12-fold higher F/E ratio than the remaining “Ordinary” subgroup. All members of this Positive Outlier subgroup showed F/E ratios less than 1.0 (mean, 0.36). The mean ratio for the Ordinary subgroup was 4.4, suggesting that if part of the body’s F comes from a pool of E, as suggested by others [45,46], then this pool is greatly reduced in the high-Hair-F, Positive Outlier subgroup and might represent a suboptimal reserve for normal HPA axis function.

Previous studies in blood have reported that F levels are reduced both acutely during an individual TM practice session [120] and chronically, in a prospective, random assignment, active-control study of male medical students after 4 months of twice-daily practice of TM [18]. More relevant to the current study, lower urinary excretion rates of F were found in a previous study of long-term (avg. 8.5 y) TM-practitioner college students compared with non-meditating controls [16]. That study also reported evidence for two classes of F excreters. The excessively high-F class, termed the “sodium-loaders”, showed markedly higher urinary excretion rates of not only F but also sodium, other minerals, and (during waking hours) vanillylmandelic acid (VMA), a metabolite of norepinephrine and epinephrine, than did the remaining controls or the TM group. These extra-high-F controls also showed a higher waking/sleeping ratio of F excretion. Excretion rates of aldosterone and dehydroepiandrosterone sulfate (DS), the other steroids measured, were not different between sodium-loaders and the remaining control subjects.

None of these other measures were examined in the present study, leaving uncertainty as to whether the high Hair F Outlier group observed here is comparable to the high-F subgroup in that study, which represented 33% of controls (vs. 0% in the TM group). It may be noteworthy that an exploratory aspect of the large hair steroid study involving 1258 factory employees, mentioned earlier, identified a subclass with extra-high Hair F that represented 16.6% of the total [20], possibly matching the Positive Outlier group in the current study.

It is unknown whether the observed high-Hair-F subgroup reflects genuine variability in long-term systemic F secretion not captured by Hair E or, alternatively, arises from differences in local enzymatic conversion of F to E in the hair follicle, as suggested previously [20]. However, more recent studies have directly shown cycling between F and E on the systemic level. For example, studies using pulses of F and E that were deuterium labeled in different carbon atoms, allowing the cycling rates between F and E to be directly determined, showed that not only does cycling between F and E occur within specific organs and tissues but also on a whole-body level under conditions such as hyperinsulinemia and therapeutic steroid administration [45,46]. In the current study, the large (12-fold) difference in Hair F/Hair E ratio between the Outlier and Ordinary groups suggests the possibility of different states of glucocorticoid regulation, perhaps different set-points in the balance between the active and inactive forms. Large shifts of the F/E ratio occur in the adrenal glands because this is where most circulating catabolic steroids (glucocorticoids) as well as anabolic steroids (e.g., DHEA) are synthesized [121]. The F/E ratio should serve as a better indicator of the balance of F and E than the concentration of either F or E alone. The possibility that a subgroup of individuals with high Hair F and high Hair F/Hair E ratio may reflect a distinctive allostatic load or overload condition deserves further investigation.

### 4.6. Glucocorticoid Activity, Allostatic Load, and Energy Usage

Our initial transcriptional comparison of long-term TM practitioners and matched controls found evidence consistent with lower allostatic load and lower, or more efficient, energy consumption [14]. The present results support these findings. Not only was *SOCS3*, whose product is known to suppress full activation of the energy regulator AMPK, confirmed to be less expressed in the Old TM group but the Hair F/Hair E ratio was lower as well, suggesting lower persistent glucocorticoid activity. This may relate to the recent study of energy use in cultured fibroblasts, mentioned earlier, which showed that cortisol in the culture medium increased energy usage [12]. The authors viewed this system as an example of “cellular allostatic load” and reported that fibroblasts cultured in the presence of cortisol also had a lower Hayflick limit, i.e., more rapid aging. The authors concluded that cellular allostatic load increases energy consumption and contributes to more rapid aging [12]. This finding, along with other research, has led to a “brain–body energy conservation” model of aging [122]. This model is consistent with other research indicating that biological age undergoes a rapid increase in response to diverse forms of stress but that this change in biological age is reversed following recovery from stress [123]. The transcriptional part of our study was conducted in blood cells (PBMCs) which, according to the hair steroid results, likely experienced chronically elevated levels of cortisol, making the comparison with this study of fibroblasts in culture a reasonable one.

Other evidence that TM practice is capable of reversing allostatic load or overload comes from studies of post-traumatic stress disorder (PTSD). A systematic review and meta-analysis of 61 studies using meditation techniques to treat PTSD reports that all four categories of techniques that have been studied produced significant benefits [124]. Analysis of the 16 studies employing TM found this technique to have a large effect size (Hedges’s g = −1.13) in reducing PTSD symptoms compared to moderate effect sizes with the three other types of meditation.

Finally, another recent article highlights the central but complex role of glucocorticoids in adaptation to stress and discusses the possibility of reversing the long-lasting effects of allostatic load contributing to disease and aging [125]. The authors elaborate on the role of the two receptors for glucocorticoids—GR and MR—in the brain and how their sequence, level, and duration of activation appear to underlie physiological, psychological, and behavioral effects, including effects on cognitive functions. These observations do not contradict the notion of age reversal by correcting effects of stress, alluded to in an earlier paragraph, and may help to understand the wide range of effects observed.

### 4.7. Limitations

This study lacked a suitable placebo, i.e., an active-control condition, for either the 12-year, Young-TM or the 40-year, Old-TM group. The study also was conducted at a single research site and at a single time point, i.e., without a baseline or known challenge condition. Further, the number of participants was relatively small and group sizes were sometimes unequal. Restricted resources prevented transcriptional analysis in the Young TM group. Where differences in paired outcomes were observed, these design features limit causal inferences. It is primarily the result of pre-existing, short-term RCTs of TM in the areas studied that, when coupled with the present results, provide a basis for proposing a causal role of long-term meditation practice.

## 5. Summary and Conclusions

The main findings of this study are consistent with the role of meditation in reducing effects of stress and aging. These are: (1) reduced expression of specific aging-related genes, including *SOCS3*, importantly related to stress and energy metabolism; (2) benefits for cognitive functions that decline with age; (3) reduced level of the Hair F/Hair E ratio and Hair F in the Young TM and the Old TM subgroups, suggesting more optimal and more stable HPA axis regulation. The prior existence of short-term RCTs and other studies whose results align with the present observations make a causal role for meditation likely.

These results, when combined with other evidence from the parent study (see [14,22]), are consistent with the conclusion that long-term practice of TM technologies can reduce or reverse allostatic load. This conclusion is supported by other studies, such as those on their efficacy in treating PTSD [124]. It is also consistent with results from a study in fibroblasts in which cellular allostatic load produced by elevated glucocorticoid level was found to cause a shift from glycolysis to oxidative phosphorylation as the main energy supply [12]. Early research on TM indicates it may foster the reverse shift, i.e., from oxidative phosphorylation to glycolysis during meditation (see, for review, [78]). Results in the parent study suggest that such a shift may pertain outside of meditation as well. From this knowledge and the known ability of SOCS3 to reduce the activity of AMPK [64], a key regulator of mitochondrial energy metabolism [65,66], a deeper understanding of the role of energetics in the benefits of meditation may be emerging [13,126].

## Figures and Tables

**Figure 1 biomolecules-15-00317-f001:**
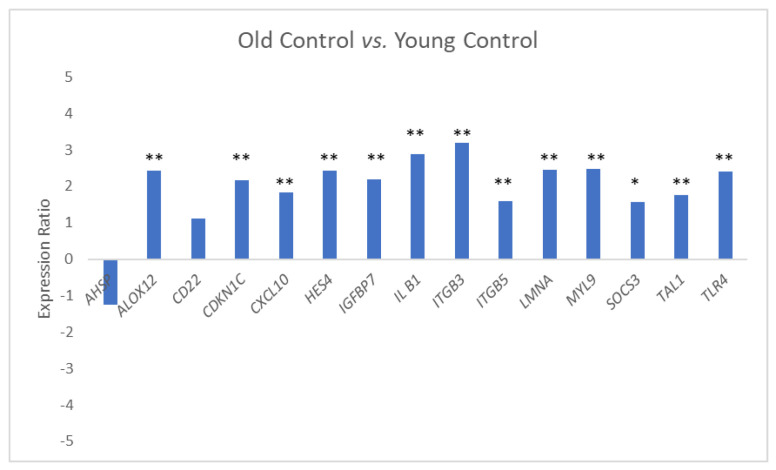
Expression ratio and statistical significance for the qPCR comparison of expression of 15 selected genes in the Old Control group (OC, *n* = 21) relative to the Young Control group (YC, *n* = 19). * *p* ≤ 0.10; ** *p* ≤ 0.05.

**Figure 2 biomolecules-15-00317-f002:**
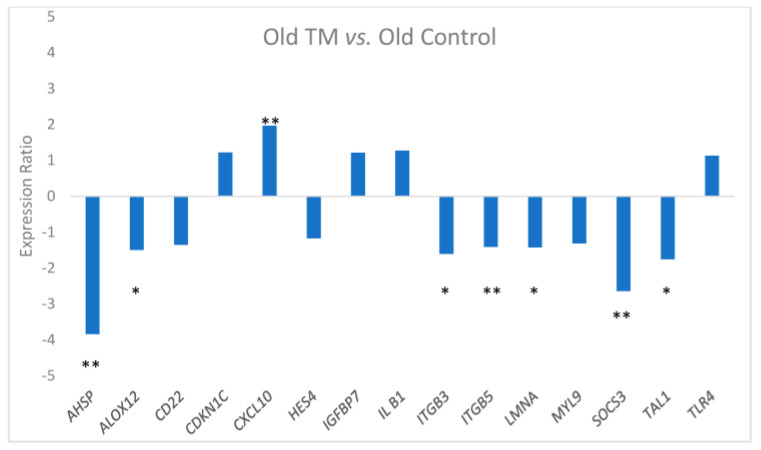
Expression ratio and statistical significance for the qPCR comparison of expression of 15 selected genes in the Old TM group (OTM, *n* = 23) relative to the Old Control group (OC, *n* = 21). (Some of these data appeared in the parent study) [14]. * *p* ≤ 0.10; ** *p* ≤ 0.05.

**Figure 3 biomolecules-15-00317-f003:**
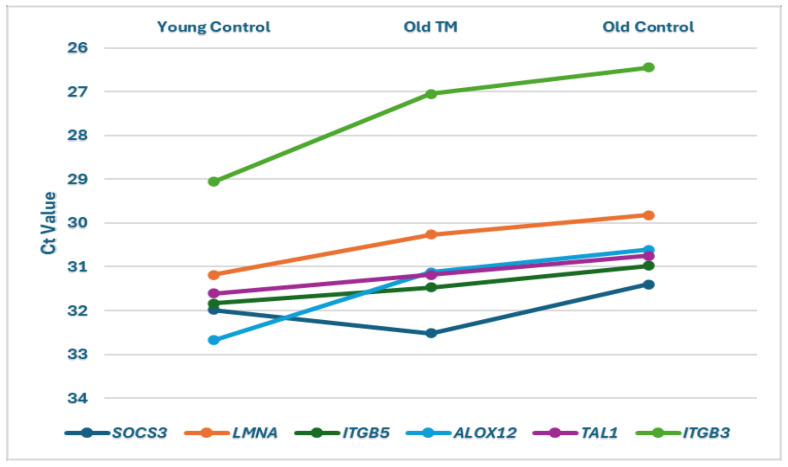
Comparison of average threshold cycle values (Ct) for 6 genes in the Young Control group (YC, *n* = 19), Old TM group (OTM, *n* = 23), and Old Control group (OC, *n* = 21).

**Figure 4 biomolecules-15-00317-f004:**
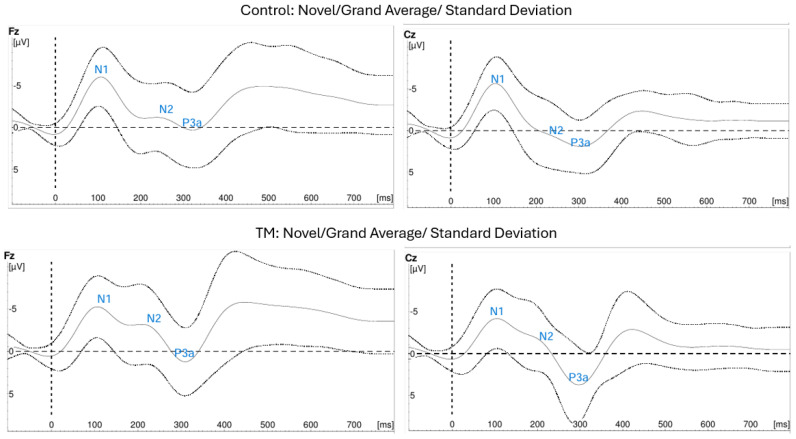
ERP Tracings showing N2 and P3a from Novel Stimuli.

**Figure 5 biomolecules-15-00317-f005:**
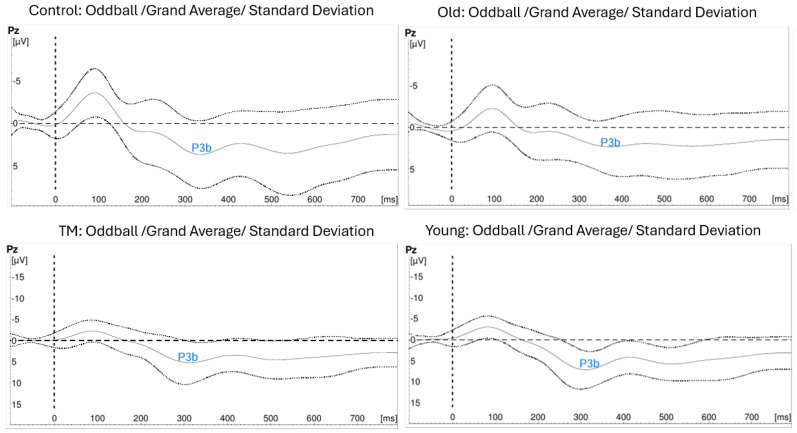
ERP Tracings showing P3b from Oddball Stimuli.

**Figure 6 biomolecules-15-00317-f006:**
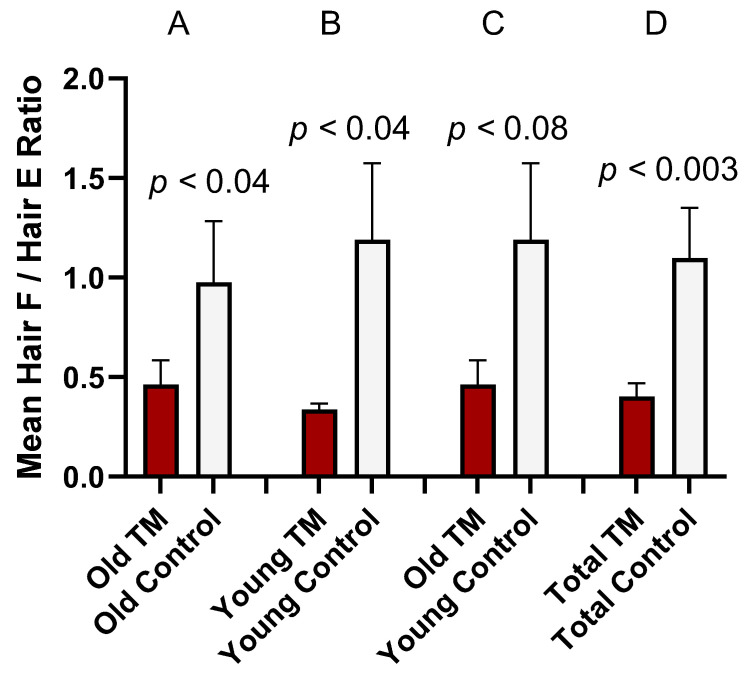
Comparison of Hair F/Hair E Ratios in TM and Control Subgroups. Statistical comparisons were made with the non-parametric Mann–Whitney U test. (A) Old TM vs. Old Control, *Z* = 2.10, *p* < 0.04, Glass’s *delta* = 0.36; (B) Young TM vs. Young Control, *Z* = 2.10, *p* < 0.04, Glass’s *delta* = 0.41; (C) Old TM vs. Young Control, *Z* = 1.77, *p* < 0.08, Glass’s *delta* = 0.35; (D) Total TM vs. Total Control, *Z* = 2.98, *p* < 0.003, Glass’s *delta* = 0.38. Error bars represent SE.

**Figure 7 biomolecules-15-00317-f007:**
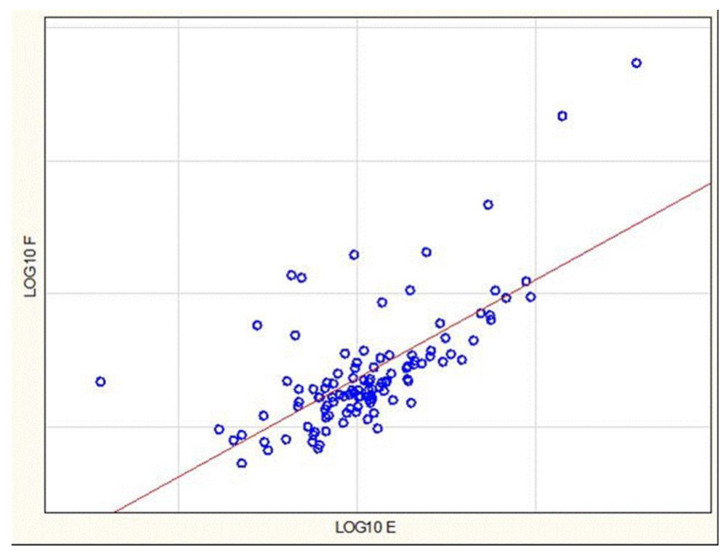
Scatterplot of Log_10_ Hair F Values vs. Log_10_ Hair E Values for all Study Participants.

**Table 1 biomolecules-15-00317-t001:** Overview of the Subject Characteristics in Each Group Comparison.

	Subject Characteristics	Young Controls	Young TM	Old Controls	Old TM
Gene Expression Analyses	N (males)	19 (8)	N/A	21 (9)	23 (14)
	Age (mean ± SD)	24.05 ± 3.06	N/A	62.43 ± 4.69	64.2 ± 4.2
	Months TM (mean ± SD)	N/A	N/A	N/A	477 ± 55
	Months TM-Sidhi (mean ± SD)	N/A	N/A	N/A	399 ± 42
Cognitive Performance Analyses	N (males)	25 (13)	30 (15)	22 (10)	26 (17)
	Age (mean ± SD)	24.2 ± 3.0	24.4 ± 2.86	62.7 ± 4.45	64.8 ± 3.5
	Months TM (mean ± SD)	N/A	146.5 ± 64.8	N/A	474.3 ± 33.21
	Months TM-Sidhi (mean ± SD)	N/A	47.1 ± 42.8	N/A	401.3 ± 27.0
Hair Glucocorticoid Analyses	N (males)	29 (15)	26 (18)	23 (10)	30 (18)
	Age (mean ± SD)	24.1 ± 2.9	24.0 ± 2.6	62.4 ± 4.7	63.8 ± 3.5
	Months TM (mean ± SD)	N/A	157.3 ± 60.1	N/A	478.8 ± 35.5
	Months TM-Sidhi (mean ± SD)	N/A	49.7 ± 39.2	N/A	399.6 ± 37.4

N/A, not applicable.

**Table 2 biomolecules-15-00317-t002:** Demographics for Participants in EEG Recording.

Subject Characteristics	Young Controls	Young TM	Old Controls	Old TM
N (males)	25 (13)	30 (15)	22 (10)	23 (14)
Age (mean ± SD)	24.2 ± 3.0	24.4 ± 2.86	62.7 ± 4.45	64.8 ± 3.5
Vegetarians	2	12	2	15
Smokers	12	13	5	0
Drinkers	8	12	5	2
Moderate Exercise	18	23	18	23
Professional, Skilled Work	3	7	14	17
Retired	0	0	6	10
Student	22	23	0	2
High School Only	6	4	5	3
College Degree	19	26	17	20
Right-Handed	23	28	19	20
Loss of Consciousness (Injury)	2	2	3	4
Non-English-Speaking 1st 20 Years	0	7	0	3
Married	2	1	12	13

**Table 3 biomolecules-15-00317-t003:** Expression Ratios and *p*-Values for Possible Anti-Aging Effect on Gene Expression.

Gene	Expression Ratio of Old Control/Young Control	*p*-Value	Expression Ratio of Old Control/Old TM	*p*-Value	Expression Ratio of Old TM/Young Control	*p*-Value
*SOCS3*	1.6	0.079	2.15	0.001	1.35 ^a^	0.219
*ITGB5*	1.56	0.009	1.38	0.022	1.13	0.518
*TAL1*	1.65	0.006	1.25	0.076	1.33	0.262
*ITGB3*	2.95	0.001	1.44	0.06	2.05	0.002
*LMNA*	2.33	0.001	1.3	0.074	1.8	0.009
*ALOX12*	2.28	0.001	1.33	0.086	1.71	0.074

^a^ Old TM was insignificantly lower than Young Control.

**Table 4 biomolecules-15-00317-t004:** Two-Factor MANOVA Comparison of N2, P3a, and P3b Latencies by TM Status and Age Group.

Comparison	N	ERP	Mean ± SEM (ms)	*F*-Statistic and Effect Size
**TM vs. Control**				
TM	41	N2	226.2 (4.2)	***F* (3,61) = 4.4, *p* = 0.007, d = 0.84**
		P3a	322.8 (4.0)	
		P3b	331.5 (6.0)	
Control	42	N2	247.8 (4.9)	
		P3a	342.6 (7.4)	
		P3b	361.4 (9.8)	
**Young vs. Old**				
Young	39	N2	217.7 (4.7)	***F* (3,62) = 10.9, *p* < 0.001, d = 0.96**
		P3a	310.4 (4.0)	
		P3b	331.5 (6.1)	
Old	44	N2	256.9 (5.1)	
		P3a	355.6 (6.5)	
		P3b	366.3 (8.5)	

**Table 5 biomolecules-15-00317-t005:** Individual Subgroup ANOVAs of N2, P3a, P3b Latencies: Old TM vs. Old Control, and Old TM vs. Young Control.

	N	ERP	Mean ± SEM (ms)	*F*-Statistic and Effect Size
**Old TM vs. Old Control**				
Old TM	22	N2	244.1 (3.3)	***F*(1,42) = 6.4, *p* = 0.015**, *η*2 = 0.13
		P3a	340.4 (4.9)	***F*(1,37) = 5.5, *p* = 0.024, ***η*2 = 0.13
		P3b	345.8 (6.7)	***F*(1,38) = 5.6, *p* = 0.023, ***η*2 = 0.13
Old Control	22	N2	269.7 (5.8)	
		P3a	373.5 (7.3)	
		P3b	394.5 (11.0)	
**Old TM vs. Young Control**				
Old TM	22	N2	244.1 (3.3)	*F*(1,42) = 3.6, *p* = 0.07, *η*2 = 0.08
		P3a	340.4 (4.9)	***F*(1,40) = 5.4, *p* = 0.024**, *η*2 = 0.12
		P3b	345.8 (6.7)	*F*(1,39) < 1.0, ns, *η*2 = 0.005
Young Control	22	N2	226.0 (5.6)	
		P3a	316.0 (4.9)	
		P3b	339.5 (6.9)	

Note: Significant differences are bolded for easy identification. Note 2: *η*2 stands for eta squared. It is a measure of effect size that indicates the proportion of variance in a dependent variable that is associated with different groups. Small effect size is *η*2 = 0.01, medium effect size is *η*2 = 0.06, and a large effect size is *η*2 = 0.14.

**Table 6 biomolecules-15-00317-t006:** Two-Factor ANOVA Comparison of BIS Scores: TM Status and Age Group.

Comparison	*N*	Mean ± SE	Significance Test	*P* Value, Effect Size (Eta Squared)
**TM Status**			***F* (1,92) = 9.46**	**<0.001,** ***η*2 = 0.11**
TM	**55**	4.63 ± 0.226		
Control	45	3.62 ± 0.238		
**Age Group**			***F* (1,92) = 12.81**	**<0.001,** ***η*2 = 0.12**
Young	**52**	4.71 ± 0.224		
Old	48	3.54 ± 0.240		

**Table 7 biomolecules-15-00317-t007:** Individual Subgroup ANOVAs of BIS Scores: Old TM vs. Old Control, and Old TM vs. Young Control.

	N	Mean ± SEM	*F*-Statistic and Effect Size (Eta Squared)	
**Old TM vs. Old Control**			***F* (1,46) = 1.9, *p* = 0.002, *η*2= 0.19**	
Old TM	26	4.2 (1.3)		
Old Control	22	2.9 (1.4)		
**Old TM vs. Young Control**			*F* (1,47) < 1.0, ns, *η*2 < 0.001	
Old TM	26	4.2 (1.3)		
Young Control	23	4.3 (2.0)		

**Table 8 biomolecules-15-00317-t008:** Glucocorticoid Comparisons in the Positive Outlier and Ordinary Groups.

	Positive Outlier Group Mean (SE)	Ordinary Group Mean (SE)	Mann–Whitney Z	*p*-Value
Hair F (pg/mg)	29.6 (9.42)	3.33 (0.383)	−4.53	6 × 10 ^−6^
Hair E (pg/mg)	20.57 (9.71)	8.82 (0.727)	−0.989	0.32
Hair F/Hair E	4.35 (0.64)	0.361 (0.015)	−5.18	<1 × 10 ^−6^

## Data Availability

The original contributions presented in this study are included in the article/Appendix A.

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
