# Peer review of "Possible Anti-Aging and Anti-Stress Effects of Long-Term Transcendental Meditation Practice: Differences in Gene Expression, EEG Correlates of Cognitive Function, and Hair Steroids"

_biomolecules, 2025, doi:10.3390/biom15030317_

Round 1
Reviewer 1 Report
Comments and Suggestions for Authors
Review:
General evaluation: The study evaluated effects of long-term meditation (TM) in older and younger adults on several genetic, endocrinological, neurophysiological and cognitive parameters. The study compared older controls with a matched TM group to assess effects of TM, a young control with an older control group to assess aging effects and younger controls with older TM group to assess the anti-aging effects of meditation. This is a comprehensive, timely and interesting study that sheds light on positive effects of meditation on mental and biological aging. The study is well conducted and follows an interdisciplinary approach. On the downside, however, there are some problems with the manuscript that diminish my enthusiasm. There are some methodological shortcomings, and the result section is confusing as the data analysis strategy is unclear. The group design was not completely balanced and there were no measures of biological markers in the young TM group. For each parameters different groups were compared. The discussion contains several sections that are related to gene expression whereas other findings are underrepresented. The discussion should be shortened and rewritten to make it more concise. Thus, there are critical points that should be considered before the manuscript can be published. Please see below for specific comments.
The authors conducted three group comparisons to assess the TM effect in the older group (old TM vs. old controls), aging effects (young controls vs. old controls) and the “anti-aging” effects by comparing young controls vs. old TM. While the first two comparisons are plausible, the last one is problematic due to potential confounding variables that have not been controlled. In my opinion, the effect of anti-aging could be better assessed by contrasting the age effect as a difference Old TM – Old control and Old TM – young control.
The introduction is rather confusing and includes paragraphs not relevant for the study (e.g. CTRA), whereas other aspects such as cognitive aspects or importance of the gene overexpression as well as the hair parameters were not sufficiently outlined to formulate precise hypotheses.
Further sociodemographic and lifestyle parameters of the participants are needed. For example, information about education, occupation, retirement, physical and cognitive activity, social components, or personality (if available).
The behavioral data of the cognitive tasks are missing. Please provide reaction times and accuracy for each group and conduct statistical analysis. Furthermore, the methodology of the BIS is unclear. What is the advantage of the measure? How it can be meaningfully interpreted when so many different parameters are simply added.
The ERPs were not presented. Thus, the group differences in the morphology of the ERP components in the different task cannot be evaluated. Also, it would be important to analyze the amplitude of the P3b and earlier ERP components. P3a was mentioned but no further considered. Also, no topography of the EEG signals was included.
Generally, more information about the hair parameters are needed (Hair F, Hair E, F/E-ratio). The absolute concentrations (non-logarithmic) of hair cortisol in pg/mg for each group is missing. How many participants were excluded due to extreme values? Why did the authors combine the Old TM and the Young groups? The rationale for building the subgroups 1 and 2 is unclear. Fig. 6 is not fully clear. What do the histograms show? Are there the normally distributed values of log10 F and E? This is unclear. Please note that logarithmic measures are unitless and the units should be removed. Please include all parameters of the regression analysis (e.g., R-square, beta).
At the beginning of the discussion some sentences about the aim of the study would be helpful and whether the hypotheses of the study were confirmed or not.
The discussion of gene expression is extraordinarily inflated. I recommend moving some of the previous evidence to the introduction and only discussing the aspects relevant to the study at the end and radically shortening the discussion. In addition, the numbering of the paragraphs is not correct.
Discussion of the P3b latency delay in aging and cortical thickness is less relevant in this context as P3b reflects primarily timing of cognitive processing. Also, the stimulus evaluation approach of the P3b is largely outdated as the latency is frequently longer than the response. Newer literature regarding functional properties of the P3b should be considered.
Discussion of BIS is insufficient. It has been mentioned that BIS included 9 EEG parameters, but no further details of the data were presented here. There should be a clear statement, what the data show or the authors may select some relevant EEG parameters and analyze the data transparently.
The main results of hair glucocorticoids need to be shortly described before other studies are discussed. This section is apparently too long and should be rewritten. Please include some of the critical points in the section "Limitations".
Reviewer 2 Report
Comments and Suggestions for Authors
This study examined the potential anti-aging and anti-stress effects of long-term Transcendental Meditation (TM) practice. The study analyses differences in gene expression, cognitive function through EEG, and hair steroid levels between long-term TM practitioners and age-matched non-meditators. Key findings suggest reduced biological aging and allostatic load in TM practitioners, supported by lower expression of stress-related genes, better cognitive performance, and lower levels of stress hormones. These results align with previous research, indicating that TM may contribute to healthier aging through improved energy metabolism and stress resilience.
This is an interesting and important study that adds to the growing knowledge base regarding the impact and mechanisms of action behind the health-related benefits of TM.
Major point
As it stands, the existing description of the study and associated study groups is confusing. The description of the experimental groups in this study could be significantly improved to enhance clarity and accessibility. Consistent and standardised terminology should be used throughout the text to clearly label the groups, such as "Young TM Practitioners" or "Old Controls." These labels should be explicitly defined at their first use and consistently applied in all discussions and figures to avoid confusion. To provide an overview, a comprehensive table summarising key details of all groups should be included. This table could list information such as group names, sample sizes, age ranges, mean ages, gender distribution, and TM practice durations, ensuring readers can easily reference the essential characteristics of each group.
Additionally, a CONSORT-style flow diagram would greatly improve the reader’s understanding of the study design. This diagram could map the progression of participants through the study, including recruitment, allocation into specific groups, data collection steps, and any attrition or exclusions. These improvements would also facilitate replication and critical evaluation of the study.
Minor points
Abstract methods: line 20-23
The abstract listing the different experimental groups is quite confusing.
Materials and Methods: line 114-115
The description of the groups in the methods section is clearer – I recommend adopting this approach for the abstract.
Results
The tables need to be streamlined aesthetically and justified left where relevant e.g., Table 3, column 1.
Figures 4, 5 and 7. I feel that the numbers (n) of each group should be represented in this graph, either on the graph itself or the figure legend. This is especially important since the numbers are not equal across the groups.
Discussion
Line 426-427: please clarify for the reader the broad function of these genes. For example, you say “lower expression of 13 of the 15 selected genes in the Young Control group than in the Old Control group”. I understand that you have already listed and discussed these genes in the manuscript prior to the discussion but it is helpful for the reader if you provide a braid description, as you do for point 3, where you say that these genes are associated with stress and the ageing process.
Line 442 – larger font size for the words “In one…”. Similar font size issues in line 449.
Line 461 – CVD is not defined as cardiovascular disease in the text and is written twice.
Line 483 - This is consonant with the recent recognition of mitochondrial energy……. I presume the authors mean “consistent”?
Line 487 – which specific stress reduction interventions does this study refer to?
Line 520 – additional space.
Line 522 – Randomised Controlled Trials (RCT). This phrase should be abbreviated in all subsequent use, including line 683 – please amend through the manuscript.
Line 668 – a supporting reference is required here.
Line 674-675 – I’m not sure that is appropriate to refer to subjective spiritual characteristics or phenomena (e.g., enlightenment) in this type of study report.
Line 776 (Limitations) – I believe that the authors should point out that studying PBMC is not necessarily reflective of organ or issue health. Furthermore, although intriguing, I think that the authors should be careful associating or inferring links between cellular lab-based studies using cultured fibroblasts with entire organisms. It is always tempting to make these links but I believe that researchers have a duty to point out these facts to readers. I realise that the authors have made some statements about causality in the manuscript but I think they should be more explicit.
Round 2
Reviewer 1 Report
Comments and Suggestions for Authors
The authors have done a good job and considered most of my comments and made some additions and corrections. I only have minor comments: The socio-demographic characteristics of the subjects are important and should be placed in the main article instead of the supplementary material. The same is true for the ERPs. However, ERPs should not be presented separately for each group, but should be overlaid according to the statistical analyses and group comparisons and included in the main text. What do the EXG channels represent? Are they averaged eye movements? In general, the methodology of EEG analysis and functional interpretation could be better described and I would recommend making the manuscript a bit more stringent and concise to enhance its readability and quality.
Author Response
Replies to reviewer 1, round 2
The authors again thank the reviewer for useful and constructive comments. Our responses to individual comments are listed below. All additions to the revised MS are shown in red typeface.
Reviewer’s Comments
- The socio-demographic characteristics of the subjects are important and should be placed in the main article instead of the supplementary material
Reply: Now inserted on page 5 (in Methods) and referenced in first paragraph of the EEG results section on page 11.
- The same is true for the ERPs. However, ERPs should not be presented separately for each group, but should be overlaid according to the statistical analyses and group comparisons and included in the main text.
Reply: Now done (Figure 4, page 12 and Figure 5, page 14). Great improvement.
- What do the EXG channels represent? Are they averaged eye movements?
Reply: EXG channels are the ear recordings, used for reference. They are no longer shown.
- In general, the methodology of EEG analysis and functional interpretation could be better described and I would recommend making the manuscript a bit more stringent and concise to enhance its readability and quality.
Reply: Descriptions of EEG methods and functional interpretation were clarified, with attention to the Introduction, Methods, Results, and Discussion sections. Tightening of the prose throughout the MS resulted in a reduction of nearly 500 words.
Reviewer 2 Report
Comments and Suggestions for Authors
Thank you for your answering my questions and making the various amendments to your manuscript. Good luck with your future work.
Author Response
Replies to reviewer 2, round 2
The authors again thank the reviewer for useful and constructive comments. Your review was most helpful!